# LARGE-SCALE OPTIMAL TRANSPORT AND MAPPING ESTIMATION

**Vivien Seguy**
Kyoto University
Graduate School of Informatics
`vivien.seguy@iip.ist.i.kyoto-u.ac.jp`

**Bharath Bhushan Damodaran**
Université de Bretagne Sud
IRISA, UMR 6074, CNRS
`bharath-bhushan.damodaran@irisa.fr`

**Rémi Flamary**
Université Côte dAzur
Lagrange, UMR 7293, CNRS, OCA
`remi.flamary@unice.fr`

**Nicolas Courty**
Université de Bretagne Sud
IRISA, UMR 6074, CNRS
`courty@univ-ubs.fr`

**Antoine Rolet**
Kyoto University
Graduate School of Informatics
`antoine.rolet@iip.ist.i.kyoto-u.ac.jp`

**Mathieu Blondel**
NTT Communication Science Laboratories
`mathieu@mblondel.org`

## ABSTRACT

This paper presents a novel two-step approach for the fundamental problem of learning an optimal map from one distribution to another. First, we learn an optimal transport (OT) plan, which can be thought as a one-to-many map between the two distributions. To that end, we propose a stochastic dual approach of regularized OT, and show empirically that it scales better than a recent related approach when the amount of samples is very large. Second, we estimate a *Monge map* as a deep neural network learned by approximating the barycentric projection of the previously-obtained OT plan. This parameterization allows generalization of the mapping outside the support of the input measure. We prove two theoretical stability results of regularized OT which show that our estimations converge to the OT plan and *Monge map* between the underlying continuous measures. We showcase our proposed approach on two applications: domain adaptation and generative modeling.

## 1 INTRODUCTION

**Mapping one distribution to another** Given two random variables $X$ and $Y$ taking values in $\mathcal{X}$ and $\mathcal{Y}$ respectively, the problem of finding a map $f$ such that $f(X)$ and $Y$ have the same distribution, denoted $f(X) \sim Y$ henceforth, finds applications in many areas. For instance, in domain adaptation, given a source dataset and a target dataset with different distributions, the use of a mapping to align the source and target distributions is a natural formulation (Gopalan et al., 2011) since theory has shown that generalization depends on the similarity between the two distributions (Ben-David et al., 2010). Current state-of-the-art methods for computing generative models such as generative adversarial networks (Goodfellow et al., 2014), generative moments matching networks (Li et al., 2015) or variational auto encoders (Kingma & Welling, 2013) also rely on finding $f$ such that $f(X) \sim Y$. In this setting, the latent variable $X$ is often chosen as a continuous random variable, such as a Gaussian distribution, and $Y$ is a discrete distribution of real data, e.g. the ImageNet dataset. By learning a map $f$, sampling from the generative model boils down to simply drawing a sample from $X$ and then applying $f$ to that sample.

**Mapping with optimality** Among the potentially many maps $f$ verifying $f(X) \sim Y$, it may be of interest to find a map which satisfies some optimality criterion. Given a cost of moving mass from one point to another, one would naturally look for a map which minimizes the total cost of transporting the mass from $X$ to $Y$. This is the original formulation of Monge (1781), which initiated

the development of the optimal transport (OT) theory. Such *optimal maps* can be useful in numerous applications such as color transfer (Ferradans et al., 2014), shape matching (Su et al., 2015), data assimilation (Reich, 2011; 2013), or Bayesian inference (Moselhy & Marzouk, 2012). In small dimension and for some specific costs, multi-scale approaches (Mérigot, 2011) or dynamic formulations (Evans & Gangbo, 1999; Benamou & Brenier, 2000; Papadakis et al., 2014; Solomon et al., 2014) can be used to compute optimal maps, but these approaches become intractable in higher dimension as they are based on space discretization. Furthermore, maps veryfiying $f(X) \sim Y$ might not exist, for instance when $X$ is a constant but not $Y$. Still, one would like to find optimal maps between distributions at least approximately. The modern approach to OT relaxes the Monge problem by optimizing over plans, i.e. distributions over the product space $\mathcal{X} \times \mathcal{Y}$, rather than maps, casting the OT problem as a linear program which is always feasible and easier to solve. However, even with specialized algorithms such as the network simplex, solving that linear program takes $O(n^3 \log n)$ time, where $n$ is the size of the discrete distribution (measure) support.

**Large-scale OT** Recently, Cuturi (2013) showed that introducing entropic regularization into the OT problem turns its dual into an easier optimization problem which can be solved using the Sinkhorn algorithm. However, the Sinkhorn algorithm does not scale well to measures supported on a large number of samples, since each of its iterations has an $\mathcal{O}(n^2)$ complexity. In addition, the Sinkhorn algorithm cannot handle continuous probability measures. To address these issues, two recent works proposed to optimize variations of the dual OT problem through stochastic gradient methods. Genevay et al. (2016) proposed to optimize a "semi-dual" objective function. However, their approach still requires $\mathcal{O}(n)$ operations per iteration and hence only scales moderately w.r.t. the size of the input measures. Arjovsky et al. (2017) proposed a formulation that is specific to the so-called 1-Wasserstein distance (unregularized OT using the Euclidean distance as a cost function). This formulation has a simpler dual form with a single variable which can be parameterized as a neural network. This approach scales better to very large datasets and handles continuous measures, enabling the use of OT as a loss for learning a generative model. However, a drawback of that formulation is that the dual variable has to satisfy the non-trivial constraint of being a Lipschitz function. As a workaround, Arjovsky et al. (2017) proposed to use weight clipping between updates of the neural network parameters. However, this makes unclear whether the learned generative model is truly optimized in an OT sense. Besides these limitations, these works only focus on the computation of the OT objective and do not address the problem of finding an optimal map between two distributions.

**Contributions** We present a novel two-step approach for learning an *optimal map* $f$ that satisfies $f(X) \sim Y$. First, we compute an optimal transport plan, which can be thought as a one-to-many map between the two distributions. To that end, we propose a new simple dual stochastic gradient algorithm for solving regularized OT which scales well with the size of the input measures. We provide numerical evidence that our approach converges faster than semi-dual approaches considered in (Genevay et al., 2016). Second, we learn an *optimal map* (also referred to as a *Monge map*) as a neural network by approximating the barycentric projection of the OT plan obtained in the first step. Parameterization of this map with a neural network allows efficient learning and provides generalization outside the support of the input measure. Fig. 1 provides a 2D example showing the computed map between a Gaussian measure and a discrete measure and the resulting density estimation. On the theoretical side, we prove the convergence of regularized optimal plans (resp. barycentric projections of regularized optimal plans) to the optimal plan (resp. Monge map) between the underlying continuous measures from which data are sampled. We demonstrate our approach on domain adaptation and generative modeling.

*Notations:* We denote $\mathcal{X}$ and $\mathcal{Y}$ some complete metric spaces. In most applications, these are Euclidean spaces. We denote random variables such as $X$ or $Y$ as capital letters. We use $X \sim Y$ to say that $X$ and $Y$ have the same distribution, and also $X \sim \mu$ to say that $X$ is distributed according to the probability measure $\mu$. Supp($\mu$) refers to the support of $\mu$, a subset of $\mathcal{X}$, which is also the set of values which $X \sim \mu$ can take. Given $X \sim \mu$ and a map $f$ defined on Supp($\mu$), $f \# \mu$ is the probability distribution of $f(X)$. We say that a measure is continuous when it admits a density w.r.t. the Lebesgues measure. We denote id the identity map.

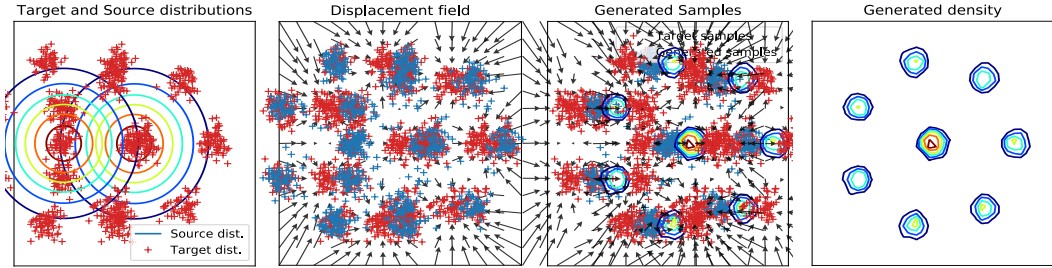

Figure 1: Example of estimated optimal map between a continuous Gaussian distribution (colored level sets) and a multi-modal discrete measure (red +). (left) Continuous source and discrete target distributions. (center left) displacement field of the estimated optimal map: each arrow is proportional to $f(x_i) - x_i$ where $(x_i)$ is a uniform discrete grid. (center right) Generated samples obtained by sampling from the source distribution and applying our estimated *Monge map* $f$. (right) Level sets of the resulting density (approximated as a 2D histogram over $10^6$ samples).

## 2 BACKGROUND ON OPTIMAL TRANSPORT

**The Monge Problem** Consider a cost function $c : (x, y) \in \mathcal{X} \times \mathcal{Y} \mapsto c(x, y) \in \mathbb{R}^+$, and two random variables $X \sim \mu$ and $Y \sim \nu$ taking values in $\mathcal{X}$ and $\mathcal{Y}$ respectively. The Monge problem (Monge, 1781) consists in finding a map $f : \mathcal{X} \to \mathcal{Y}$ which transports the mass from $\mu$ to $\nu$ while minimizing the mass transportation cost,

$$\inf_f \ \mathbb{E}_{X \sim \mu} \left[ c(X, f(X)) \right] \ \text{subject to } f(X) \sim Y. \tag{1}$$

Monge originally considered the cost $c(x, y) = \|x - y\|_2$, but in the present article we refer to the Monge problem as Problem (1) for any cost $c$. When $\mu$ is a discrete measure, a map $f$ satisfying the constraint may not exist: if $\mu$ is supported on a single point, no such map exists as soon as $\nu$ is not supported on a single point. In that case, the Monge problem is not feasible. However, when $\mathcal{X} = \mathcal{Y} = \mathbb{R}^d$, $\mu$ admits a density and $c$ is the squared Euclidean distance, an important result by Brenier (1991) states that the Monge problem is feasible and that the infinum of Problem (1) is attained. The existence and uniqueness of *Monge maps*, also referred to as *optimal maps*, was later generalized to more general costs (e.g. strictly convex and super-linear) by several authors. With the notable exception of the Gaussian to Gaussian case which has a close form affine solution, computation of *Monge maps* remains an open problem for measures supported on high-dimensional spaces.

**Kantorovich Relaxation** In order to make Problem (1) always feasible, Kantorovich (1942) relaxed the Monge problem by casting Problem (1) into a minimization over couplings $(X, Y) \sim \pi$ rather than the set of maps, where $\pi$ should have marginals equals to $\mu$ and $\nu$,

$$\inf_\pi \ \mathbb{E}_{(X,Y) \sim \pi} \left[ c(X, Y) \right] \ \text{subject to } X \sim \mu, \ Y \sim \nu. \tag{2}$$

Concretely, this relaxation allows mass at a given point $x \in \text{Supp}(\mu)$ to be transported to several locations $y \in \text{Supp}(\nu)$, while the Monge problem would send the whole mass at $x$ to a unique location $f(x)$. This relaxed formulation is a linear program, which can be solved by specialized algorithms such as the network simplex when considering discrete measures. However, current implementations of this algorithm have a super-cubic complexity in the size of the support of $\mu$ and $\nu$, preventing wider use of OT in large-scale settings.

**Regularized OT** OT regularization was introduced by Cuturi (2013) in order to speed up the computation of OT. Regularization is achieved by adding a negative-entropy penalty $R$ (defined in Eq. (5)) to the primal variable $\pi$ of Problem (2),

$$\inf_\pi \ \mathbb{E}_{(X,Y) \sim \pi} \left[ c(X, Y) \right] + \varepsilon R(\pi) \ \text{subject to } X \sim \mu, \ Y \sim \nu. \tag{3}$$

Besides efficient computation through the Sinkhorn algorithm, regularization also makes the OT distance differentiable everywhere w.r.t. the weights of the input measures (Blondel et al., 2018), whereas OT is differentiable only almost everywhere. We also consider the $L^2$ regularization introduced by Dessein et al. (2016), whose computation is found to be more stable since there is no

exponential term causing overflow. As highlighted by Blondel et al. (2018), adding an entropy or squared $L^2$ norm regularization term to the primal problem (3) makes the dual problem an unconstrained maximization problem. We use this dual formulation in the next section to propose an efficient stochastic gradient algorithm.

# 3 LARGE-SCALE OPTIMAL TRANSPORT

By considering the dual of the regularized OT problem, we first show that stochastic gradient ascent can be used to maximize the resulting concave objective. A close form for the primal solution $\pi$ of Problem (3) can then be obtained by using first-order optimality conditions.

## 3.1 DUAL STOCHASTIC APPROACH

**OT dual** Let $X \sim \mu$ and $Y \sim \nu$. The Kantorovich duality provides the following dual of the OT problem (2),

$$\sup_{u \in \mathcal{C}(\mathcal{X}), v \in \mathcal{C}(\mathcal{Y})} \mathbb{E}_{(X,Y) \sim \mu \times \nu} \left[ u(X) + v(Y) \right] \quad \text{subject to } u(x) + v(y) \leqslant c(x,y) \text{ for all } (x,y). \quad (4)$$

This dual formulation suggests that stochastic gradient methods can be used to maximize the objective of Problem (4) by sampling batches from the independant coupling $\mu \times \nu$. However there is no easy way to fulfill the constraint on $u$ and $v$ along gradient iterations. This motivates considering regularized optimal transport.

**Regularized OT dual** The hard constraint in Eq. (4) can be relaxed by regularizing the primal problem (2) with a strictly convex regularizer $R$ as detailed in (Blondel et al., 2018). In the present paper, we consider both entropy regularization $R_e$ used in (Cuturi, 2013; Genevay et al., 2016) and $L^2$ regularization $R_{L^2}$,

$$R_e(\pi) \overset{\text{def.}}{=} \int_{\mathcal{X} \times \mathcal{Y}} \left( \ln \left( \frac{d\pi(x,y)}{d\mu(x)d\nu(y)} \right) - 1 \right) d\pi(x,y), \quad R_{L^2}(\pi) \overset{\text{def.}}{=} \int_{\mathcal{X} \times \mathcal{Y}} \left( \frac{d\pi(x,y)}{d\mu(x)d\nu(y)} \right)^2 d\mu(x)d\nu(y).$$
$$(5)$$

where $\frac{d\pi(x,y)}{d\mu(x)d\nu(y)}$ is the density, i.e. the Radon-Nikodym derivative, of $\pi$ w.r.t. $\mu \times \nu$. When $\mu$ and $\nu$ are discrete, and so is $\pi$, the integrals are replaced by sums. The dual of the regularized OT problems can be obtained through the Fenchel-Rockafellar's duality theorem,

$$\sup_{u,v} \mathbb{E}_{(X,Y) \sim \mu \times \nu} \left[ u(X) + v(Y) + F_\varepsilon(u(X), v(Y)) \right], \quad (6)$$

$$\text{where } F_\varepsilon(u(x), v(y)) = \begin{cases} -\varepsilon e^{\frac{1}{\varepsilon}(u(x)+v(y)-c(x,y))} & \text{(entropy reg.)} \\ -\frac{1}{4\varepsilon}(u(x)+v(y)-c(x,y))_+^2 & (L^2 \text{ reg.}) \end{cases} . \quad (7)$$

Compared to Problem (4), the constraint $u(x) + v(y) \leqslant c(x,y)$ has been relaxed and is now enforced smoothly through a penalty term $F_\varepsilon(u(x), v(y))$ which is concave w.r.t. $(u,v)$. Although we derive formula and perform experiments w.r.t. entropy and $L^2$ regularizations, any strictly convex regularizer which is decomposable, i.e. which can be written $R(\pi) = \sum_{ij} R_{ij}(\pi_{ij})$ (in the discrete case), gives rise to a dual problem of the form Eq. (6), and the proposed algorithms can be adapted.

**Primal-Dual relationship** In order to recover the solution $\pi^\varepsilon$ of the regularized primal problem (3), we can use the first-order optimality conditions of the Fenchel-Rockafellar's duality theorem,

$$d\pi^\varepsilon(x,y) = H_\varepsilon(x,y)d\mu(x)d\nu(y) \text{ where } H_\varepsilon(x,y) = \begin{cases} e^{\frac{u(x)}{\varepsilon}} e^{-\frac{c(x,y)}{\varepsilon}} e^{\frac{v(y)}{\varepsilon}} & \text{(entropy reg.)} \\ \frac{1}{2\varepsilon}(u(x)+v(y)-c(x,y))_+ & (L^2 \text{ reg.}) \end{cases} . \quad (8)$$

**Algorithm** The relaxed dual (6) is an unconstrained concave problem which can be maximized through stochastic gradient methods by sampling batches from $\mu \times \nu$. When $\mu$ is discrete, i.e. $\mu = \sum_{i=1}^{n} a_i \delta_{x_i}$, the dual variable $u$ is a $n$-dimensional vector over which we carry the optimization, where $u(x_i) \overset{\text{def.}}{=} u_i$. When $\mu$ has a density, $u$ is a function on $\mathcal{X}$ which has to be parameterized in order to carry optimization. We thus consider deep neural networks for their ability to approximate

---

**Algorithm 1** Stochastic OT computation

1: **Inputs:** input measures $\mu$, $\nu$; cost function $c$; batch size $p$; learning rate $\gamma$.
2: Discrete case: $\mu = \sum_i a_i \delta_{x_i}$ and $u$ is a finite vector: $u(x_i) \stackrel{\text{def}}{=} u_i$ (similarly for $\nu$ and $v$)
3: Continuous case: $\mu$ is a continuous measure and $u$ is a neural network (similarly for $\nu$ and $v$)
   $\nabla$ indicates the gradient w.r.t. the parameters
4: **while** not converged **do**
5:     sample a batch $(x_1, \cdots, x_p)$ from $\mu$
6:     sample a batch $(y_1, \cdots, y_p)$ from $\nu$
7:     update $u \leftarrow u + \gamma \sum_{ij} \nabla u(x_i) + \partial_u F_\varepsilon(u(x_i), v(y_j)) \nabla u(x_i)$
8:     update $v \leftarrow v + \gamma \sum_{ij} \nabla v(y_j) + \partial_v F_\varepsilon(u(x_i), v(y_j)) \nabla v(y_j)$
9: **end while**

---

general functions. Genevay et al. (2016) used the same stochastic dual maximization approach to compute the regularized OT objective in the continuous-continuous setting. The difference lies in their pamaterization of the dual variables as kernel expansions, while we decide to use deep neural networks. Using a neural network for parameterizing a continuous dual variable was done also by Arjovsky et al. (2017). The same discussion also stands for the second dual variable $v$. Our stochastic gradient algorithm is detailed in Alg. 1.

**Convergence rates and computational cost comparison.** We first discuss convergence rates in the discrete-discrete setting (i.e. both measures are discrete), where the problem is convex, while parameterization of dual variables as neural networks in the semi-discrete or continuous-continuous settings make the problem non-convex. Because the dual (6) is not strongly convex, full-gradient descent converges at a rate of $\mathcal{O}(1/k)$, where $k$ is the iteration number. SGD with a decreasing step size converges at the inferior rate of $\mathcal{O}(1/\sqrt{k})$ (Nemirovski et al., 2009), but with a $\mathcal{O}(1)$ cost per iteration. The two rates can be interpolated when using mini-batches, at the cost of $\mathcal{O}(p^2)$ per iteration, where $p$ is the mini-batch size. In contrast, Genevay et al. (2016) considered a semi-dual objective of the form $\mathbb{E}_{X \sim \mu} [u(X) + G_\varepsilon(u(X))]$, with a cost per iteration which is now $\mathcal{O}(n)$ due to the computation of the gradient of $G_\varepsilon$. Because that objective is not strongly convex either, SGD converges at the same $O(1/\sqrt{k})$ rate, up to problem-specific constants. As noted by Genevay et al. (2016), this rate can be improved to $\mathcal{O}(1/k)$ while maintaining the same iteration cost, by using stochastic average gradient (SAG) method (Schmidt et al., 2017). However, SAG requires to store past stochastic gradients, which can be problematic in a large-scale setting.

In the semi-discrete setting (i.e. one measure is discrete and the other is continuous), SGD on the semi-dual objective proposed by Genevay et al. (2016) also converges at a rate of $\mathcal{O}(1/\sqrt{k})$, whereas we only know that Alg. 1 converges to a stationary point in this non-convex case.

In the continuous-continuous setting (i.e. both measures are continuous), Genevay et al. (2016) proposed to represent the dual variables as kernel expansions. A disadvantage of their approach, however, is the $\mathcal{O}(k^2)$ cost per iteration. In contrast, our approach represents dual variables as neural networks. While non-convex, our approach preserves a $\mathcal{O}(p^2)$ cost per iteration. This parameterization with neural networks was also used by Arjovsky et al. (2017) who maximized the 1-Wasserstein dual-objective function $\mathbb{E}_{(X,Y) \sim \mu \times \nu} [u(X) - u(Y)]$. Their algorithm is hence very similar to ours, with the same complexity $\mathcal{O}(p^2)$ per iteration. The main difference is that they had to constrain $u$ to be a Lipschitz function and hence relied of weight clipping in-between gradient updates. The proposed algorithm is capable of computing the regularized OT objective and optimal plans between empirical measures supported on arbitrary large numbers of samples. In statistical machine learning, one aims at estimating the underlying continuous distribution from which empirical observations have been sampled. In the context of optimal transport, one would like to approximate the true (non-regularized) optimal plan between the underlying measures. The next section states theoretical guarantees regarding this problem.

## 3.2 CONVERGENCE OF REGULARIZED OT PLANS

Consider discrete probability measures $\mu_n = \sum_{i=1}^n a_i \delta_{x_i} \in P(\mathcal{X})$ and $\nu_n = \sum_{j=1}^n b_j \delta_{y_j} \in P(\mathcal{Y})$. Analysis of entropy-regularized linear programs (Cominetti & San Martín, 1994) shows that the

solution $\pi_n^\varepsilon$ of the entropy-regularized problem (3) converges exponentially fast to a solution $\pi_n$ of the non-regularized OT problem (2). Also, a result about stability of optimal transport (Villani, 2008)[Theorem 5.20] states that, if $\mu_n \to \mu$ and $\nu_n \to \nu$ weakly, then a sequence $(\pi_n)$ of optimal transport plans between $\mu_n$ and $\nu_n$ converges weakly to a solution $\pi$ of the OT problem between $\mu$ and $\nu$. We can thus write,

$$\lim_{n\to\infty} \lim_{\varepsilon\to 0} \pi_n^\varepsilon = \pi. \tag{9}$$

A more refined result consists in establishing the weak convergence of $\pi_n^\varepsilon$ to $\pi$ when $(n, \varepsilon)$ jointly converge to $(\infty, 0)$. This is the result of the following theorem which states a stability property of entropy-regularized plans (proof in the Appendix).

**Theorem 1.** *Let $\mu \in P(\mathcal{X})$ and $\nu \in P(\mathcal{Y})$ where $\mathcal{X}$ and $\mathcal{Y}$ are complete metric spaces. Let $\mu_n = \sum_{i=1}^n a_i \delta_{x_i}$ and $\nu_n = \sum_{j=1}^n b_j \delta_{y_j}$ be discrete probability measures which converge weakly to $\mu$ and $\nu$ respectively, and let $(\varepsilon_n)$ a sequence of non-negative real numbers converging to $0$ sufficiently fast. Assume the cost $c$ is continuous on $\mathcal{X} \times \mathcal{Y}$ and finite. Let $\pi_n^{\varepsilon_n}$ the solution of the entropy-regularized OT problem (3) between $\mu_n$ and $\nu_n$. Then, up to extraction of a subsequence, $(\pi_n^{\varepsilon_n})$ converges weakly to the solution $\pi$ of the OT problem (2) between $\mu$ and $\nu$,*

$$\pi_n^{\varepsilon_n} \to \pi \quad weakly. \tag{10}$$

Keeping the analogy with statistical machine learning, this result is an analog to the universal consistency property of a learning method. In most applications, we consider empirical measures and $n$ is fixed, so that regularization, besides enabling dual stochastic approach, may also help learn the optimal plan between the underlying continuous measures.

So far, we have derived an algorithm for computing the regularized OT objective and regularized optimal plans regardless of $\mu$ and $\nu$ being discrete or continuous. The OT objective has been used successfully as a loss in machine learning (Montavon et al., 2015; Frogner et al., 2015; Rolet et al., 2016; 2018; Arjovsky et al., 2017; Courty et al., 2017a), whereas the use of optimal plans has straightforward applications in logistics, as well as economy (Kantorovich, 1942; Carlier, 2012) or computer graphics (Bonneel et al., 2011). In numerous applications however, we often need mappings rather than joint distributions. This is all the more motivated since Brenier (1991) proved that when the source measure is continuous, the optimal transport plan is actually induced by a map. Assuming that available data samples are sampled from some underlying continuous distributions, finding the Monge map between these continuous measures rather than a discrete optimal plan between discrete measures is essential in machine learning applications. Hence in the next section, we investigate how to recover an optimal map, i.e. find an approximate solution to the Monge problem (1), from regularized optimal plans.

## 4 Optimal Mapping Estimations

A map can be obtained from a solution to the OT problem (2) or regularized OT problem (3) through the computation of its barycentric projection. Indeed, a solution $\pi$ of Problem (2) or (3) between a source measure $\mu$ and a target measure $\nu$ is, identifying the plan $\pi$ with its density w.r.t. a reference measure, a function $\pi : (x, y) \in \mathcal{X} \times \mathcal{Y} \mapsto \mathbb{R}^+$ which can be seen as a weighted one-to-many map, i.e. $\pi$ sends $x$ to each location $y \in \text{Supp}(\nu)$ where $\pi(x, y) > 0$. A map can then be obtained by simply averaging over these $y$ according to the weights $\pi(x, y)$.

**Definition 1.** *(Barycentric projection) Let $\pi$ be a solution of the OT problem (2) or regularized OT problem (3). The barycentric projection $\bar{\pi}$ w.r.t. a convex cost $d : \mathcal{Y} \times \mathcal{Y} \to \mathbb{R}^+$ is defined as,*

$$\bar{\pi}(x) = \arg\min_z \mathbb{E}_{Y \sim \pi(\cdot|x)} [d(z, Y)]. \tag{11}$$

In the special case $d(x, y) = \|x - y\|_2^2$, Eq. (11) has the close form solution $\bar{\pi}(x) = \mathbb{E}_{Y \sim \pi(\cdot|x)} [Y]$, which is equal to $\bar{\pi} = \frac{\pi \mathbf{y}^t}{a}$ in a discrete setting with $\mathbf{y} = (y_1, \cdots, y_n)$ and $a$ the weights of $\mu$. Moreover, for the specific squared Euclidean cost $c(x, y) = \|x - y\|_2^2$, the barycentric projection $\bar{\pi}$ is an *optimal map* (Ambrosio et al., 2006)[Theorem 12.4.4], i.e. $\bar{\pi}$ is a solution to the Monge problem (1) between the source measure $\mu$ and the target measure $\bar{\pi}\#\mu$. Hence the barycentric projection w.r.t. the squared Euclidean cost is often used as a simple way to recover optimal maps from optimal transport plans (Reich, 2013; Wang et al., 2013; Ferradans et al., 2014; Seguy & Cuturi, 2015).

---

**Algorithm 2** Optimal map learning with SGD

---

**Inputs:** input measures $\mu$, $\nu$; cost function $c$; dual optimal variables $u$ and $v$; map $f_\theta$ parameterized as a deep NN; batch size $n$; learning rate $\gamma$.
**while** not converged **do**
    sample a batch $(x_1, \cdots, x_n)$ from $\mu$
    sample a batch $(y_1, \cdots, y_n)$ from $\nu$
    update $\theta \leftarrow \theta - \gamma \sum_{ij} H_\varepsilon(x_i, y_j) \nabla_\theta d(y_j, f_\theta(x_i))$
**end while**

---

Formula (11) provides a pointwise value of the barycentric projection. When $\mu$ is discrete, this means that we only have mapping estimations for a finite number of points. In order to define a map which is defined everywhere, we parameterize the barycentric projection as a deep neural network. We show in the next paragraph how to efficiently learn its parameters.

**Optimal map learning** An estimation $f$ of the barycentric projection of a regularized plan $\pi^\varepsilon$ *which generalizes outside the support* of $\mu$ can be obtained by learning a deep neural network which minimizes the following objective w.r.t. the parameters $\theta$,

$$\mathbb{E}_{X \sim \mu} \left[ \mathbb{E}_{Y \sim \pi^\varepsilon(\cdot|X)} \left[ d(Y, f_\theta(X)) \right] \right] = \mathbb{E}_{(X,Y) \sim \pi^\varepsilon} \left[ d(Y, f_\theta(X)) \right]$$
$$= \mathbb{E}_{(X,Y) \sim \mu \times \nu} \left[ d(Y, f_\theta(X)) H_\varepsilon(X, Y) \right]. \tag{12}$$

When $d(x, y) = \|x - y\|^2$, the last term in Eq. (12) is simply a weighted sum of squared errors, with possibly an infinite number of terms whenever $\mu$ or $\nu$ are continuous. We propose to minimize the objective (12) by stochastic gradient descent, which provides the simple Algorithm 2. The OT problem being symmetric, we can also compute the opposite barycentric projection $g$ w.r.t. a cost $d : \mathcal{X} \times \mathcal{X} \to \mathbb{R}^+$ by minimizing $\mathbb{E}_{(X,Y) \sim \mu \times \nu} \left[ d(g(Y), X) H_\varepsilon(X, Y) \right]$.

However, unless the plan $\pi$ is induced by a map, the averaging process results in having the image of the source measure by $\bar{\pi}$ only approximately equal to the target measure $\nu$. Still, when the size of discrete measure is large and the regularization is small, we show in the next paragraph that 1) the barycentric projection of a regularized OT plan is close to the Monge map between the underlying continuous measures (Theorem 2) and 2) the image of the source measure by this barycentric projection should be close to the target measure $\nu$ (Corollary 1).

**Theoretical guarantees** As stated earlier, when $\mathcal{X} = \mathcal{Y}$ and $c(x, y) = \|x - y\|_2^2$, Brenier (1991) proved that when the source measure $\mu$ is continuous, there exists a solution to the Monge problem (1). This result was generalized to more general cost functions, see (Villani, 2008)[Corollary 9.3] for details. In that case, the plan $\pi$ between $\mu$ and $\nu$ is written as $(\text{id}, f)\#\mu$ where $f$ is the Monge map. Now considering discrete measures $\mu_n$ and $\nu_n$ which converge to $\mu$ (continuous) and $\nu$ respectively, we have proved in Theorem 1 that $\pi_n^\varepsilon$ converges weakly to $\pi = (\text{id}, f)\#\mu$ when $(n, \varepsilon) \to (\infty, 0)$. The next theorem, proved in the Appendix, shows that the barycentric projection $\bar{\pi}_n^\varepsilon$ also converges weakly to the true Monge map between $\mu$ and $\nu$, justifying our approach.

**Theorem 2.** *Let $\mu$ be a continuous probability measure on $\mathbb{R}^d$, and $\nu$ an arbitrary probability measure on $\mathbb{R}^d$ and $c$ a cost function satisfying (Villani, 2008)[Corollary 9.3]. Let $\mu_n = \frac{1}{n} \sum_{i=1}^n \delta_{x_i}$ and $\nu_n = \frac{1}{n} \sum_{j=1}^n \delta_{y_j}$ converging weakly to $\mu$ and $\nu$ respectively. Assume that the OT solution $\pi_n$ of Problem (2) between $\mu_n$ and $\nu_n$ is unique for all $n$. Let $(\varepsilon_n)$ a sequence of non-negative real numbers converging sufficiently fast to 0 and $\bar{\pi}_n^{\varepsilon_n}$ the barycentric projection w.r.t. the convex cost $d = c$ of the solution $\pi_n^{\varepsilon_n}$ of the entropy-regularized OT problem (3). Then, up to extraction of a subsequence,*

$$(\text{id}, \bar{\pi}_n^{\varepsilon_n})\#\mu_n \to (\text{id}, f)\#\mu \quad weakly, \tag{13}$$

*where $f$ is the solution of the Monge problem (1) between $\mu$ and $\nu$.*

This theorem shows that our estimated barycentric projection is close to an optimal map between the underlying continuous measures for $n$ big and $\varepsilon$ small. The following corollary confirms the intuition that the image of the source measure by this map converges to the underlying target measure.

**Corollary 1.** *With the same assumptions as above, $\bar{\pi}_n^{\varepsilon_n}\#\mu_n \to \nu \quad weakly$.*

In terms of random variables, the last equation states that if $X_n \sim \mu_n$ and $Y \sim \nu$, then $\bar{\pi}_n^{\varepsilon_n}(X_n)$ converges in distribution to $Y$.

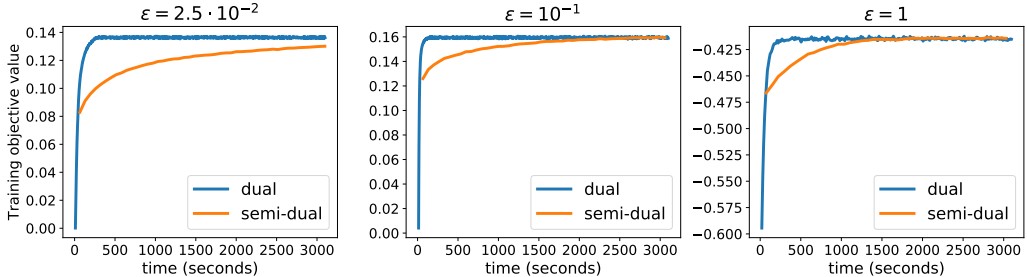

Figure 2: Convergence plots of the the Stochastic Dual Algorithm 1 against a stochastic semi-dual implementation (adapted from (Genevay et al., 2016): we use SGD instead of SAG), for several entropy-regularization values. Learning rates are $\{5., 20., 20.\}$ and batch sizes $\{1024, 500, 100\}$ respectively and are taken the same for the dual and semi-dual methods.

These theoretical results show that our estimated Monge map can thus be used to perform domain adaptation by mapping a source dataset to a target dataset, as well as perform generative modeling by mapping a continuous measure to a target discrete dataset. We demontrate this in the following section.

## 5 NUMERICAL EXPERIMENTS

### 5.1 DUAL VS SEMI-DUAL SPEED COMPARISONS

We start by evaluating the training time of our dual stochastic algorithm 1 against a stochastic semi-dual approach similar to (Genevay et al., 2016). In the semi-dual approach, one of the dual variable is eliminated and is computed in close form. However, this computation has $\mathcal{O}(n)$ complexity where $n$ is the size of the target measure $\nu$. We compute the regularized OT objective with both methods on a spectral transfer problem, which is related to the color transfer problem (Reinhard et al., 2001; Pitié et al., 2007), but where images are multispectral, *i.e.* they share a finer sampling of the light wavelength. We take two $500 \times 500$ images from the CAVE dataset (Yasuma et al., 2010) that have 31 spectral bands. As such, the optimal transport problem is computed on two empirical distributions of 250000 samples in $\mathbb{R}^{31}$ on which we consider the squared Euclidean ground cost $c$. The timing evolution of train losses are reported in Figure 2 for three different regularization values $\varepsilon = \{0.025, 0.1, 1.\}$. In the three cases, one can observe that convergence of our proposed dual algorithm is much faster.

### 5.2 LARGE SCALE DOMAIN ADAPTATION

We apply here our computation framework on an unsupervised domain adaptation (DA) task, for which optimal transport has shown to perform well on small scale datasets (Courty et al., 2017b; Perrot et al., 2016; Courty et al., 2014). This restriction is mainly due to the fact that those works only consider the primal formulation of the OT problem. Our goal here is not to compete with the state-of-the-art methods in domain adaptation but to assess that our formulation allows to scale optimal transport based domain adaptation (OTDA) to large datasets. OTDA is illustrated in Fig. 3 and follows two steps: 1) learn an optimal map between the source and target distribution, 2) map the source samples and train a classifier on them in the target domain. Our formulation also allows to use any differentiable ground cost $c$ while (Courty et al., 2017b) was limited to the squared Euclidean distance.

**Datasets** We consider the three cross-domain digit image datasets MNIST (Lecun et al., 1998), USPS, and SVHN (Netzer et al., 2011), which have 10 classes each. For the adaptation between MNIST and USPS, we use 60000 samples in the MNIST domain and 9298 samples in USPS domain. MNIST images are resized to the same resolution as USPS ones $(16 \times 16)$. For the adaptation between SVHN and MNIST, we use 73212 samples in the SVHN domain and 60000 samples in the MNIST domain. MNIST images are zero-padded to reach the same resolution as SVHN $(32 \times 32)$ and extended to three channels to match SVHN image sizes. The labels in the target domain are

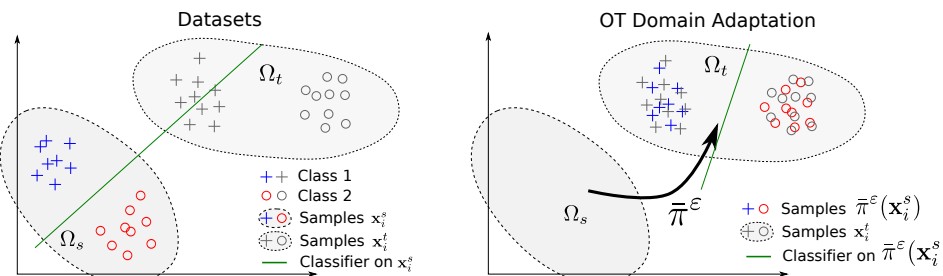

Figure 3: Illustration of the OT Domain Adaptation method adapted from (Courty et al., 2017b). Source samples are mapped to the target set through the barycentric projection $\bar{\pi}^\varepsilon$. A classifier is then learned on the mapped source samples.

Table 1: Results (accuracy in %) on domain adaptation among MNIST, USPS and SVHN datasets with entropy ($R_e$) and L2 ($R_{L^2}$) regularizations. *Source only* refers to 1-NN classification between source and target samples without adaptation.

| Method | MNIST→ USPS | USPS→ MNIST | SVHN → MNIST |
|---|---|---|---|
| Source only | 73.47 | 36.97 | 54.33 |
| Bar. proj. OT | 57.75 | 52.46 | intractable |
| Bar. proj. OT with $R_e$ | 68.75 | 57.35 | intractable |
| Bar. proj. Alg. 1 with $R_e$ | 68.84 | 57.55 | 58.87 |
| Bar. proj. Alg. 1 with $R_{L^2}$ | 67.8 | 57.47 | 60.56 |
| Monge map Alg. 1+2 with $R_e$ | **77.92** | 60.02 | 61.11 |
| Monge map Alg. 1+2 with $R_{L^2}$ | 72.61 | **60.50** | **62.88** |

withheld during the adaptation. In the experiment, we consider the adaptation in three directions: MNIST → USPS, USPS → MNIST, and SVHN → MNIST.

**Methods and experimental setup** Our goal is to demonstrate the potential of the proposed method in large-scale settings. Adaptation performance is evaluated using a 1-nearest neighbor (1-NN) classifier, since it has the advantage of being parameter free and allows better assessment of the quality of the adapted representation, as discussed in (Courty et al., 2017b). In all experiments, we consider the 1-NN classification as a baseline, where labeled neighbors are searched in the source domain and the accuracy is computed on target data. We compare our approach to previous OTDA methods where an optimal map is obtained through the discrete barycentric projection of either an optimal plan (computed with the network simplex algorithm[1]) or an entropy-regularized optimal plan (computed with the Sinkhorn algorithm (Cuturi, 2013)), whenever their computation is tractable. Note that these methods do not provide out-of-sample mapping. In all experiments, the ground cost $c$ is the squared Euclidean distance and the barycentric projection is computed w.r.t. that cost. We learn the Monge map of our proposed approach with either entropy or L2 regularizations. Regarding the adaptation between SVHN and MNIST, we extract deep features by learning a modified LeNet architecture on the source data and extracting the 100-dimensional features output by the top hidden layer. Adaptation is performed on those features. We report for all the methods the best accuracy over the hyperparameters on the target dataset. While this setting is unrealistic in a practical DA application, it is widely used in the DA community (Long et al., 2013) and our goal is here to investigate the relative performances of large-scale OTDA in a fair setting.

**Hyper-parameters and learning rate** The value for the regularization parameter is set in $\{5, 2, 0.9, 0.5, 0.1, 0.05, 0.01\}$. Adam optimizer with batch size 1000 is used to optimize the network. The learning rate is varied in $\{2, 0.9, 0.1, 0.01, 0.001, 0.0001\}$. The learned Monge map $f$ in Alg. 2 is parameterized as a neural network with two fully-connected hidden layers ($d \rightarrow 200 \rightarrow 500 \rightarrow d$) and ReLU activations, and the weights are optimized using the Adam optimizer with learning rate equal to $10^{-4}$ and batch size equal to 1000. For the Sinkhorn algorithm, regularization value is chosen from $\{0.01, 0.1, 0.5, 0.9, 2.0, 5.0, 10.0\}$.

---

[1]http://liris.cnrs.fr/~nbonneel/FastTransport/

Figure 4: Samples generated by our optimal generator learned through Algorithms 1 and 2.

**Results** Results are reported in Table 1. In all cases, our proposed approach outperforms previous OTDA algorithms. On MNIST→USPS, previous OTDA methods perform worse than using directly source labels, whereas our method leads to successful adaptation results with 20% and 10% accuracy points over OT and regularized OT methods respectively. On USPS→MNIST, all three algorithms lead to successful adaptation results, but our method achieves the highest adaptation results. Finally, on the challenging large-scale adaptation task SVHN→MNIST, only our method is able to handle the whole datasets, and outperforms the source only results.

Comparing the results between the barycentric projection and estimated Monge map illustrates that learning a parametric mapping provides some kind of regularization, and improves the performance.

### 5.3 GENERATIVE OPTIMAL TRANSPORT (GOT)

**Approach** Corollary 1 shows that when the support of the discrete measures $\mu$ and $\nu$ is large and the regularization $\varepsilon$ is small, then we have approximately $\bar{\pi}^\varepsilon \# \mu = \nu$. This observation motivates the use of our Monge map estimation as a generator between an arbitrary continuous measure $\mu$ and a discrete measure $\nu$ representing the discrete distribution of some dataset. We can thus obtain a generative model by first computing regularized OT through Alg. 1 between a Gaussian measure $\mu$ and a discrete dataset $\nu$ and then compute our generator with Alg. 2. This requires to have a cost function between the latent variable $X \sim \mu$ and the discrete variable $Y \sim \nu$. The property we gain compared to other generative models is that our generator is, at least approximately, an *optimal map* w.r.t. this cost. In our case, the Gaussian is taken with the same dimensionality as the discrete data and the squared Euclidean distance is used as ground cost $c$.

**Permutation-invariant MNIST** We preprocess MNIST data by rescaling grayscale values in $[-1, 1]$. We run Alg. 1 and Alg. 2 where $\mu$ is a Gaussian whose mean and covariance are taken equal to the empirical mean and covariance of the preprocessed MNIST dataset; we have observed that this makes the learning easier. The target discrete measure $\nu$ is the preprocessed MNIST dataset. Permutation invariance means that we consider each grayscale $28 \times 28$ images as a 784-dimensional vector and do not rely on convolutional architectures. In Alg. 1 the dual potential $u$ is parameterized as a $(d \to 1024 \to 1024 \to 1)$ fully-connected NN with ReLU activations for each hidden layer, and the $L^2$ regularization is considered as it produced experimentally less blurring. The barycentric projection $f$ of Alg. 2 is parameterized as a $(d \to 1024 \to 1024 \to d)$ fully-connected NN with ReLU activation for each hidden layer and a $\tanh$ activation on the output layer. We display some generated samples in Fig. 4.

## 6 CONCLUSION

We proposed two original algorithms that allow for *i)* large-scale computation of regularized optimal transport *ii)* learning an optimal map that moves one probability distribution onto another (the so-called *Monge map*). To our knowledge, our approach introduces the first tractable algorithms for computing both the regularized OT objective and optimal maps in large-scale or continuous settings. We believe that these two contributions enable a wider use of optimal transport strategies in machine learning applications. Notably, we have shown how it can be used in an unsupervised

domain adaptation setting, or in generative modeling, where a Monge map acts directly as a generator. Our consistency results show that our approach is theoretically well-grounded. An interesting direction for future work is to investigate the corresponding convergence rates of the empirical regularized optimal plans. We believe this is a very complex problem since technical proofs regarding convergence rates of the empirical OT objective used e.g. in (Sriperumbudur et al., 2012; Boissard et al., 2014; Fournier & Guillin, 2015) do not extend simply to the optimal transport plans.

ACKNOWLEDGMENTS

This work benefited from the support of the project OATMIL ANR-17-CE23-0012 of the French National Research Agency (ANR). We thank the anonymous reviewers and Arthur Mensh for the careful reading and helpful comments regarding the present article.

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

# Appendix

## A  PROOFS

**Proof of Theorem 1**.

*Proof.* Let $\pi_n$ be the solution of the OT problem (2) between $\mu_n$ and $\nu_n$ which has maximum entropy. A result about stability of optimal transport (Villani, 2008)[Theorem 5.20] states that, up to extraction of a subsequence, $\pi_n$ converges weakly to a solution $\pi$ of the OT problem between $\mu$ and $\nu$ (regardless of $\pi_n$ being the solution with maximum entropy or not). We still write $(\pi_n)$ this subsequence, as well as $(\pi_n^{\varepsilon_n})$ the corresponding subsequence.
Let $g \in \mathcal{C}_b(\mathcal{X} \times \mathcal{Y})$ a bounded continuous function on $\mathcal{X} \times \mathcal{Y}$. We have,

$$\int_{\mathcal{X} \times \mathcal{Y}} g d\pi_n^{\varepsilon_n} - \int_{\mathcal{X} \times \mathcal{Y}} g d\pi = \left( \int_{\mathcal{X} \times \mathcal{Y}} g d\pi_n^{\varepsilon_n} - \int_{\mathcal{X} \times \mathcal{Y}} g d\pi_n \right) + \left( \int_{\mathcal{X} \times \mathcal{Y}} g d\pi_n - \int_{\mathcal{X} \times \mathcal{Y}} g d\pi \right)$$
(14)

The second term in the right-hand side converges to $0$ as a result of the previously mentioned stability of optimal transport (Villani, 2008)[Theorem 5.20]. We now show the convergence of the first term to $0$ when $\varepsilon_n \to 0$ sufficiently fast. We have,

$$\left| \int_{\mathcal{X} \times \mathcal{Y}} g d\pi_n^{\varepsilon_n} - \int_{\mathcal{X} \times \mathcal{Y}} g d\pi_n \right| = \left| \sum_{i=1,n \; j=1,n} g(x_i, y_j) \pi_n^{\varepsilon_n}(x_i, y_j) - \sum_{i=1,n \; j=1,n} g(x_i, y_j) \pi_n(x_i, y_j) \right|$$

$$\leqslant M_g \sum_{ij} |\pi_n^{\varepsilon_n}(x_i, y_j) - \pi_n(x_i, y_j)|$$

$$= M_g \| \pi_n^{\varepsilon_n} - \pi_n \|_{\mathbb{R}^{n \times n}, 1}$$
(15)

where $M_g$ is an upper-bound of $g$. A convergence result by Cominetti & San Martín (1994) shows that there exists positive constants (w.r.t. $\varepsilon_n$) $M_{c_n, \mu_n, \nu_n}$ and $\lambda_{c_n, \mu_n, \nu_n}$ such that,

$$\| \pi_n^{\varepsilon_n} - \pi_n \|_{\mathbb{R}^{n \times n}, 1} \leqslant M_{c_n, \mu_n, \nu_n} e^{-\frac{\lambda_{c_n, \mu_n, \nu_n}}{\varepsilon_n}}$$
(16)

where $c_n = (c(x_1, y_1), \cdots, c(x_n, y_n))$. The subscript indices indicate the dependences of each constant. Hence, we see that choosing any $(\varepsilon_n)$ such that the right-hand side of Eq. (16) tends to $0$ provides the results. In particular, we can take

$$\varepsilon_n = \frac{\lambda_{c_n, \mu_n, \nu_n}}{\ln(n M_{c_n, \mu_n, \nu_n})}$$
(17)

which suffices to have the convergence of (15) to $0$ for any bounded continuous function $g \in \mathcal{C}_b(\mathcal{X} \times \mathcal{Y})$. This proves the weak convergence of $\pi_n^{\varepsilon_n}$ to $\pi$. □

**Proof of Theorem 2**.

*Proof.* First, note that the existence of a Monge map between $\mu$ and $\nu$ follows from the absolute continuity of $\mu$ and the assumptions on the cost functions $c$ (Villani, 2008)[Corollary 9.3].
Let $g \in \mathcal{C}_l(\mathbb{R}^d \times \mathbb{R}^d)$ a Lipschitz function on $\mathbb{R}^d \times \mathbb{R}^d$. Let $\pi_n$ be the unique (by assumption) solution of the OT problem between $\mu_n$ and $\nu_n$. We have,

$$\int_{\mathbb{R}^d \times \mathbb{R}^d} g d(\mathrm{id}, \bar{\pi}_n^{\varepsilon_n}) \# \mu_n - \int_{\mathbb{R}^d \times \mathbb{R}^d} g d(\mathrm{id}, f) \# \mu = \left( \int_{\mathbb{R}^d \times \mathbb{R}^d} g d(\mathrm{id}, \bar{\pi}_n^{\varepsilon_n}) \# \mu_n - \int_{\mathbb{R}^d \times \mathbb{R}^d} g d(\mathrm{id}, \bar{\pi}_n) \# \mu_n \right)$$

$$+ \left( \int_{\mathbb{R}^d \times \mathbb{R}^d} g d(\mathrm{id}, \bar{\pi}_n) \# \mu_n - \int_{\mathbb{R}^d \times \mathbb{R}^d} g d(\mathrm{id}, f) \# \mu \right)$$
(18)

Since $\mu_n$ and $\nu_n$ are uniform discrete probability measures supported on the same number of points, we know by (Birkhoff, 1946) that the optimal transport $\pi_n$ is actually an optimal assignment $T_n$, so

that we have $\pi_n = (\text{id}, T_n)\#\mu_n$. This also implies $\bar{\pi}_n = T_n$ so that $(\text{id}, \bar{\pi}_n)\#\mu_n = (\text{id}, T_n)\#\mu_n$. Hence, the second term in the right-hand side of (18) converges to 0 as a result of the stability of optimal transport (Villani, 2008)[Theorem 5.20]. Now, we show that the first term also converges to 0 for $\varepsilon_n$ converging sufficiently fast to 0. By definition of the pushforward operator,

$$\int_{\mathbb{R}^d \times \mathbb{R}^d} g d(\text{id}, \bar{\pi}_n^{\varepsilon_n})\#\mu_n - \int_{\mathbb{R}^d \times \mathbb{R}^d} g d(\text{id}, \bar{\pi}_n)\#\mu_n = \int_{\mathbb{R}^d} g(x, \bar{\pi}_n^{\varepsilon_n}(x)d\mu_n(x) - \int_{\mathbb{R}^d} g(x, T_n(x))d\mu_n(x) \tag{19}$$

and we can bound,

$$\left| \int_{\mathbb{R}^d} g(x, \bar{\pi}_n^{\varepsilon_n}(x))d\mu_n(x) - \int_{\mathbb{R}^d} g(x, T_n(x))d\mu_n(x) \right| = \left| \frac{1}{n} \sum_{i=1}^n g(x_i, \bar{\pi}_n^{\varepsilon_n}(x_i)) - \frac{1}{n} \sum_{i=1}^n g(x_i, T_n(x_i)) \right|$$

$$\leqslant \sum_i K_g \|\bar{\pi}_n^{\varepsilon_n}(x_i) - T_n(x_i)\|_{\mathbb{R}^d, 2}$$

$$= nK_g \|\pi_n^{\varepsilon_n} Y_n - \pi_n Y_n\|_{\mathbb{R}^{n \times n}, 2}$$

$$\leqslant nK_g \|\pi_n^{\varepsilon_n} - \pi_n\|_{\mathbb{R}^{n \times n}, 2}^{1/2} \|Y_n\|_{\mathbb{R}^{n \times d}, 2}^{1/2} \tag{20}$$

where $Y_n = (y_1, \cdots, y_n)^t$ and $K_g$ is the Lipschitz constant of $g$. The first inequality follows from $g$ being Lipschitz. The next equality follows from the discrete close form of the barycentric projection. The last inequality is obtained through Cauchy-Schwartz. We can now use the same arguments as in the previous proof. A convergence result by Cominetti & San Martín (1994) shows that there exists positive constants (w.r.t. $\varepsilon_n$) $M_{c_n, \mu_n, \nu_n}$ and $\lambda_{c_n, \mu_n, \nu_n}$ such that,

$$\|\pi_n^{\varepsilon_n} - \pi_n\|_{\mathbb{R}^{n \times n}, 2}^{1/2} \leqslant M_{c_n, \mu_n, \nu_n} e^{-\frac{\lambda_{c_n, \mu_n, \nu_n}}{\varepsilon_n}} \tag{21}$$

where $c_n = (c(x_1, y_1), \cdots, c(x_n, y_n))$. The subscript indices indicate the dependences of each constant. Hence, we see that choosing any $(\varepsilon_n)$ such that (21) tends to 0 provides the results. In particular, we can take

$$\varepsilon_n = \frac{\lambda_{c_n, \mu_n, \nu_n}}{\ln(n^2 \|Y_n\|_{\mathbb{R}^{n \times d}, 2}^{1/2} M_{c_n, \mu_n, \nu_n})} \tag{22}$$

which suffices to have the convergence of (15) to 0 for Lipschitz function $g \in \mathcal{C}_l(\mathbb{R}^d \times \mathbb{R}^d)$. This proves the weak convergence of $(\text{id}, \bar{\pi}_n^{\varepsilon_n})\#\mu_n$ to $(\text{id}, f)\#\mu$. $\square$

**Proof of Corollary 1**.

*Proof.* Let $h \in \mathcal{C}_b(\mathbb{R}^d)$ a bounded continuous function. Let $g \in \mathcal{C}_b(\mathbb{R}^d \times \mathbb{R}^d)$ defined as $g : (x, y) \mapsto h(y)$. We have,

$$\int_{\mathbb{R}^d} h d\bar{\pi}_n^{\varepsilon_n} \#\mu_n - \int_{\mathbb{R}^d} h d f \#\mu = \int_{\mathbb{R}^d \times \mathbb{R}^d} g d(\text{id}, \bar{\pi}_n^{\varepsilon_n})\#\mu_n - \int_{\mathbb{R}^d \times \mathbb{R}^d} g d(\text{id}, f)\#\mu \tag{23}$$

which converges to 0 by Theorem (2). Since $f\#\mu = \nu$, this proves the corollary. $\square$

