# OpenReview forum: "Large Scale Optimal Transport and Mapping Estimation"
_ICLR.cc/2018/Conference — Accept (Poster)_

### Official Review · AnonReviewer1 · 2017-11-27
**The paper presents interestings results about consistency of learning OT/Monge maps although weak and stochastic learning algorithms able to scale, however some parts should deserve more discussion and experimental evaluation is limited.**

**Rating:** 7
**Confidence:** 3

**Review:**

Quality
The theoretical results presented in the paper appear to be correct. However, the experimental evaluation is globally limited,  hyperparameter tuning on test which is not fair.

Clarity
The paper is mostly clear, even though some parts deserve more discussion/clarification (algorithm, experimental evaluation).

Originality
The theoretical results are original, and the SGD approach is a priori original as well.

Significance
The relaxed dual formulation and OT/Monge maps convergence results are interesting and can of of interest for researchers in the area, the other aspects of the paper are limited.

Pros:
-Theoretical results on the convergence of OT/Monge maps
-Regularized formulation compatible with SGD
Cons
-Experimental evaluation limited
-The large scale aspect lacks of thorough analysis
-The paper presents 2 contributions but at then end of the day, the development of each of them appears limited

Comments:

-The weak convergence results are interesting. However, the fact that no convergence rate is given makes the result weak.
In particular, it is possible that the number of examples needed for achieving a given approximation is at least exponential.
This can be coherent with the problem of Domain Adaptation that can be NP-hard even under the co-variate shift assumption (Ben-David&Urner, ALT2012).
Then, I think that the claim of page 6 saying that Domain Adaptation can be performed "nearly optimally" has then to be rephrased.
I think that results show that the approach is theoretically justified but optimality is not here yet.

Theorem 1 is only valid for entropy-based regularizations, what is the difficulty for having a similar result with L2 regularization?

-The experimental evaluation on the running time is limited to one particular problem. If this subject is important, it would have been interesting to compare the approaches on other large scale problems and possibly with other implementations.
It is also surprising that the efficiency the L2-regularized version is not evaluated.
For a paper interesting in large scale aspects, the experimental evaluation is rather weak.

The 2 methods compared in Fig 2 reach the same objective values at convergence, but is there any particular difference in the solutions found?

-Algorithm 1 is presented without any discussion about complexity, rate of convergence. Could the authors discuss this aspect?
The presentation of this algo is a bit short and could deserve more space (in the supplementary)

-For the DA application, the considered datasets are classic but not really "large scale", anyway this is a minor remark.
The setup is not completely clear, since the approach is interesting for out of sample data, so I would expect the map to be computed on a small sample of source data, and then all source instances to be projected on target with the learned map. This point is not very clear and we do not know how many source instances are used to compute the mapping - the mapping is incomplete on this point while this is an interesting aspect of the paper: this justifies even more the large scale aspect is the algo need less examples during learning to perform similar or even better classification.
Hyperparameter tuning is another aspect that is not sufficiently precise in the experimental setup: it seems that the parameters are tuned on test (for all methods), which is not fair since target label information will not be available from a practical standpoint.

The authors claim that they did not want to compete with state of the art DA, but the approach of Perrot et al., 2016 seems to a have a similar objective and could be used as a baseline.

Experiments on generative optimal transport are interesting and probably generate more discussion/perspectives.

--
After rebuttal
--
Authors have answered to many of my comments, I think this is an interesting paper, I increase my score.

---

> ### Comment · AnonReviewer4 · 2017-12-03
> **Convergence rate**
>
> "In particular, it is possible that the number of examples needed for achieving a given approximation is at least exponential."
>
> The direct empirical Wasserstein estimator actually has a rate exponential in the input dimension (Sriperumbudur et al. 2012, On the empirical estimation of integral probability metrics, Corollary 3.5 – this is an upper bound and I don't know if there's a known matching lower bound, but I think it's relatively accepted that this is the case), so it's probably likely that this estimator's rate would be as well in the general nonparametric case.

---

> ### Author Response · Authors · 2017-12-25
> **Thank you very much for your review. Paper has been updated.**
>
> Dear reviewer,
>
> We thank you for your positive review and have updated the paper accordingly.
>
> “-The weak convergence results are interesting. However, the fact that no convergence rate is given makes the result weak. […] approximation is at least exponential.”
>
> We thank Reviewer4 for the clarification and reference. Indeed, we expect that the number of samples to achieve a given error on the OT plans grows exponentially with the dimension since it was proven in the case of the OT objective (Boissard (2011), Sriperumbudur et al. (2012), Boissard & Le Gouic (2014)), and we expect the behavior is at least as ‘bad’ for the convergence of OT plans. An interesting line of research, mentioned in conclusion of Weed and Bach (2017) is to investigate whether regularization helps improve these rates.
>
> Regarding the convergence rates of empirical OT plans, we believe this is an interesting but complex topic which deserves a study in its own right. To our knowledge, there are works proving convergence rates of the empirical OT objective (see ref. above), but none about convergence rates of OT plans.
>
> “[...]DA can be performed "nearly optimally" has then to be rephrased.”
>
> We agree and have rephrased accordingly.
>
> “Theorem 1[...], what is the difficulty for having a similar result with L2 regularization?”
>
> Our proofs rely partly on asymptotic convergence rates of entropy-reg linear programs established by Cominetti & Saint Martin (1994). To our knowledge, no extension has been obtained for L2-reg linear programs, which prevents us from adapting our proofs. Extending these results to the L2 case would be indeed of great interest.
>
> “-The experimental evaluation on the running time is limited […]. It is also surprising that the efficiency the L2-regularized version is not evaluated. […], the experimental evaluation is rather weak.”
>
> No algorithm for computing the L2-reg OT in large-scale or continuous settings have been proposed. Hence, we do not know other algorithms to compare with.
>
> We mention that our experiments are large-scale considering the OT problem. For ex., Genevay et al. (2016) considered measures supported on 20k samples, while  measures in our numerical-speed xp have 250k samples. However, we agree that more numerical-speed xps would make our proposed Alg. 1 more convincing and will add experiments.
>
> “The 2 methods compared in Fig 2 [...], is there any particular difference in the solutions found?”
>
> We performed speed-comparison experiments in the discrete setting, where the dual objective is strictly concave with a unique solution. The semi-dual objective is also strictly concave, and the dual variable solution of the semi-dual is the same as the first dual variable of the dual problem.
>
> “-Algorithm 1 is presented without any discussion about complexity, rate of convergence.”
>
> We agree and have added a paragraph “Convergence rates and computational cost comparison”.
>
> “The setup is not completely clear, since the approach is interesting for out of sample data, so I would expect the map to be computed on a small sample of source data, and then all source instances to be projected on target with the learned map. […] needs less examples during learning to perform similar or even better classification.”
>
> One of our contribution is indeed to allow out-of-sample prediction which avoids learning again a full transport map if one dataset is augmented. But learning a Monge map is a very difficult problem and one should use all the available data, which is now possible thanks to our proposed stochastic algorithms. The fact that Perrot et al. (2016) used at most 1000 samples was due to the numerical complexity of the mapping estimation alg.
>
> “Hyperparameter tuning is another aspect that is not sufficiently precise in the experimental setup: it seems that the parameters are tuned on test […].”
>
> The parameter validation tuned on test is indeed unrealistic because we have indeed not access to target samples labels in practice. Still we believe it is reasonable and fair since it allows all methods to work at their best, without relying on approximate validation that might benefit one method over another. Note that unsupervised DA validation is still an open problem: some authors perform as we did; or do validation using labels; others do more realistic but less stable techniques such as circular validation.
>
> “The authors claim that they did not want to compete with state of the art DA, but the approach of Perrot et al., 2016 seems to a have a similar objective and could be used as a baseline.”
>
> We cannot compare fairly to Perrot et al. (2016)  since they used a very small number of sample to estimate a map. But this would be a good baseline to show the importance of learning with a many samples. The method will be added to the xps very soon.
>
> Reference not in the paper:
> -Weed, Jonathan, and Francis Bach. "Sharp asymptotic and finite-sample rates of convergence of empirical measures in Wasserstein distance." arXiv

---

### Official Review · AnonReviewer2 · 2017-11-29
**An interesting paper on large scale optimal transport, though "overstating" some properties**

**Rating:** 6
**Confidence:** 3

**Review:**

The paper proves the weak convergence of the regularised OT problem to Kantorovich / Monge optimal transport problems.

I like the weak convergence results, but this is just weak convergence. It appears to be an overstatement to claim that the approach "nearly-optimally" transports one distribution to the other (Cf e.g. Conclusion). There is a penalty to pay for choosing a small epsilon -- it seems to be visible from Figure 2. Also, near-optimality would refer to some parameters being chosen in the best possible way. I do not see that from the paper. However, the weak convergence results are good.

A better result, hinting on how "optimal" this can be, would have been to guarantee that the solution to regularised OT is within f(epsilon) from the optimal one, or from within f(epsilon) from the one with a smaller epsilon (more possibilities exist). This is one of the things experimenters would really care about -- the price to pay for regularisation compared to the unknown unregularized optimum.

I also like the choice of the two regularisers and wonder whether the authors have tried to make this more general, considering other regularisations ? After all, the L2 one is just an approximation of the entropic one.

Typoes:

1- Kanthorovich -> Kantorovich (Intro)
2- Cal C <-> C (eq. 4)

---

> ### Author Response · Authors · 2017-12-25
> **Thank you very much for your review. Paper updated according to your comments.**
>
> Dear reviewer,
>
> We thank you for your positive review and relevant comments.
>
> "I like the weak convergence results, but this is just weak convergence. It appears to be an overstatement to claim that the approach "nearly-optimally" transports one distribution to the other (Cf e.g. Conclusion). There is a penalty to pay for choosing a small epsilon -- it seems to be visible from Figure 2. Also, near-optimality would refer to some parameters being chosen in the best possible way. I do not see that from the paper. However, the weak convergence results are good."
>
> Theorem 1. proves weak convergence of regularized discrete plans. This is a natural convergence for random variables (we emphasize that weak convergence is equivalent to the convergence w.r.t., for instance, the Wasserstein distance). Regarding the convergence of Monge maps (in Theorem 2), other types of convergence, such as convergence in probability, would be of great interest indeed. We may consider this problem in some future work.
>
> Using the term 'nearly-optimality' was indeed vague as we have not defined what ‘nearly’ means. We have removed this expression from the the paper. Otherwise, ‘optimal’ or ‘optimality’ refers to a solution of either the Monge problem (1), the OT problem (2), or the regularized OT problem (3).
>
> “A better result, hinting on how "optimal" this can be, would have been to guarantee that the solution to regularised OT is within f(epsilon) from the optimal one, or from within f(epsilon) from the one with a smaller epsilon (more possibilities exist). This is one of the things experimenters would really care about -- the price to pay for regularisation compared to the unknown unregularized optimum.”
>
> We can indeed consider two cases when to measure how a solution to the regularized OT (ROT) problem is ‘optimal’:
> - How close in the solution of ROT to the solution of OT w.r.t. a given norm: in the discrete case, the paper of Cominetti & Saint Martin (1994) proves asymptotic exponential convergence rate for the entropic regularization case. We are not aware of similar result for the L2 regularization, which would be of great interest and deserves a study in its own right. In the continuous-continuous case, the recent paper from Carlier et al. (2017) only provides convergence results of entropy-regularized plans.
> - How optimal is the OT objective computed with the solution of ROT: in that case various bounds about the ROT objective compared to the OT objective can be used. See for example Blondel et al. (2017) which provides bounds for both entropic and L2 regularizations.
>
> “I also like the choice of the two regularisers and wonder whether the authors have tried to make this more general, considering other regularisations ? After all, the L2 one is just an approximation of the entropic one.”
>
> This is indeed be possible. To extend our approach seamlessly, it would be sufficient that the regularizer R verifies: convexity, which ensures that the dual is well defined and unconstrained, and decomposability, which provides a dual of the form Eq. (6). More details are given in Blondel et al. (2017). We have added a small discussion about it in the main text of the updated paper, in the paragraph "Regularized OT dual".
>
> “Typoes:
> 1- Kanthorovich -> Kantorovich (Intro)
> 2- Cal C <-> C (eq. 4)”
>
> This has been corrected, thank you.
>
> References not in the paper:
> Carlier, Guillaume, et al. "Convergence of entropic schemes for optimal transport and gradient flows." SIAM Journal on Mathematical Analysis 49.2 (2017): 1385-1418.

---

### Official Review · AnonReviewer4 · 2017-12-03
**Interesting idea for expanding optimal transport estimation, but some aspects unclear**

**Rating:** 6
**Confidence:** 3

**Review:**

This paper proposes a new method for estimating optimal transport plans and maps among continuous distributions, or discrete distributions with large support size. First, the paper proposes a dual algorithm to estimate Kantorovich plans, i.e. a coupling between two input distributions minimizing a given cost function, using dual functions parameterized as neural networks. Then an algorithm is given to convert a generic plan into a Monge map, a deterministic function from one domain to the other, following the barycenter of the plan. The algorithms are shown to be consistent, and demonstrated to be more efficient than an existing semi-dual algorithm. Initial applications to domain adaptation and generative modeling are also shown.

These algorithms seem to be an improvement over the current state of the art for this problem setting, although more of a discussion of the relationship to the technique of Genevay et al. would be useful: how does your approach compare to the full-dual, continuous case of that paper if you simply replace their ball of RKHS functions with your class of deep networks?

The consistency properties are nice, though they don't provide much insight into the rate at which epsilon should be decreased with n or similar properties. The proofs are clear, and seem correct on a superficial readthrough; I have not carefully verified them.

The proofs are mainly limited in that they don't refer in any way to the class of approximating networks or the optimization algorithm, but rather only to the optimal solution. Although of course proving things about the actual outcomes of optimizing a deep network is extremely difficult, it would be helpful to have some kind of understanding of how the class of networks in use affects the solutions. In this way, your guarantees don't say much more than those of Arjovsky et al., who must assume that their "critic function" reaches the global optimum: essentially you add a regularization term, and show that as the regularization decreases it still works, but under seemingly the same kind of assumptions as Arjovsky et al.'s approach which does not add an explicit regularization term at all. Though it makes sense that your regularization might lead to a better estimator, you don't seem to have shown so either in theory or empirically.

The performance comparison to the algorithm of Genevay et al. is somewhat limited: it is only on one particular problem, with three different hyperparameter settings. Also, since Genevay et al. propose using SAG for their algorithm, it seems strange to use plain SGD; how would the results compare if you used SAG (or SAGA/etc) for both algorithms?

In discussing the domain adaptation results, you mention that the L2 regularization "works very well in practice," but don't highlight that although it slightly outperforms entropy regularization in two of the problems, it does substantially worse in the other. Do you have any guesses as to why this might be?

For generative modeling: you do have guarantees that, *if* your optimization and function parameterization can reach the global optimum, you will obtain the best map relative to the cost function. But it seems that the extent of these guarantees are comparable to those of several other generative models, including WGANs, the Sinkhorn-based models of Genevay et al. (2017, https://arxiv.org/abs/1706.00292/), or e.g. with a different loss function the MMD-based models of Li, Swersky, and Zemel (ICML 2015) / Dziugaite, Roy, and Ghahramani (UAI 2015). The different setting than the fundamental GAN-like setup of those models is intriguing, but specifying a cost function between the source and the target domains feels exceedingly unnatural compared to specifying a cost function just within one domain as in these other models.

Minor:

In (5), what is the purpose of the -1 term in R_e? It seems to just subtract a constant 1 from the regularization term.

---

> ### Author Response · Authors · 2017-12-25
> **Thank you very much for the detailed comments. Paper updated.**
>
> Dear reviewer,
>
> We thank you for your positive review and detailed comments.
>
> "how does your approach compare to the full-dual, continuous case of that paper [..]”
>
> Conceptually there is no difference. The main advantage of using NNs lies in the implementation side: using kernel expansions has a O((iteration index)^2) cost per iterations, while using NNs keeps a constant O(batch size) cost.
>
> We have added a paragraph “Convergence rates and computational cost comparison”.
>
> "The consistency properties are nice, though they don't provide much insight into the rate [..].”
>
> -For a fixed number of samples and the reg. decreasing to 0:  Cominetti & Saint Martin (1994) proved an exponential rate for the convergence of the entropy-reg. OT plans to a non-regularized OT plan. This  asymptotic result does not let infer a regularization value to achieve a given error. Building on top of these results would deserve a study in its own right.
>
> -When reg. is fixed (or 0), and the number of samples grows to inf.:  Several works study convergence rates of empirical Wasserstein distances (i.e. the OT objective between empirical measures). Boissard (2011), Sriperumbudur et al. (2012) (thanks to reviewer 4 for this ref.), Boissard & Le Gouic (2014), to name a few. However we are not aware of work addressing the same questions for the empirical OT plans (and not just the OT objective). We believe this problem is more complicated.
>
> Since our results relate to the convergence of OT plans, we believe they are new and of interest. Without them, our discussion in the introduction and experiments would not be theoretically well grounded: we could not justify that the image of a source measure through the learned Monge map approximates well the target measure, at least for some n big and eps small (Corollary 1). We understand that convergence rates are more useful and will investigate this in future work.
>
> "The proofs are mainly limited in that they don't refer in any way to the class of approximating networks [...] essentially you add a regularization term, and show that as the regularization decreases it still works, but under seemingly the same kind of assumptions as Arjovsky et al.'s approach which does not add an explicit regularization term at all. [...]"
>
> In a discrete setting, our Alg. 1 computes the exact regularized-OT since we are maximizing a concave objective without parameterizing the dual variables.
>
> Whenever the problem involves continuous measure(s), our NN parameterization only gives the exact solution when the latter belongs to this approximating class of NNs (and the global maximum is obtained).
>
> As you wrote, we do believe that this parameterization provides a “smoother” solution. But as we already have some entropic or L2 regularization in the OT problem, we find it complicated to analyze. Still, we agree that this is an interesting problem to investigate.
>
> Arjovsky et al. used indeed the same idea of deep NN  parameterization. However, their NN has to Lipschitz, which they enforce by weights clipping. This is unclear whether a NN with bounded weights can approximate any Lipschitz function. In our case, there is no restriction on the type of NNs.
>
> "The performance comparison to the algorithm of Genevay et al. is somewhat limited [...] how would the results compare if you used SAG (or SAGA/etc) for both algorithms?"
>
> We plan to add numerical-speed experiments in the paper soon.
>
> Genevay et al. used SAG in the discrete setting (but used SGD in other settings). We prefer 1) providing a unified alg. regardless of the measures being discrete or continuous, 2) proposing an alg. which fits in automatic-differentiation softwares (Tensorflow, Torch etc.), which often do not support SAG.
>
> "In discussing the domain adaptation results, you mention that the L2 regularization "works very well in practice," [...]."
>
> We have removed this sentence. It is still unclear which regularization works better in practice depending on the problem. Our only claim is that the L2 reg. is numerically more stable than the entropic one.
>
> "For generative modeling: you do have guarantees that, *if* [...] but specifying a cost function between the source and the target domains feels exceedingly unnatural […].”
>
> Indeed, most generative models focus on fitting a generator to a target distribution, without optimality criteria. Yet we believe that looking for generator which has properties can be useful for some applications. We see this experiment as a proof-of-concept that the learned Monge map can be good generator. We encourage and will consider future work where optimality of the generator (w.r.t. to a cost) is important (such as image-to-image / text-to-text translations).
>
> "In (5), what is the purpose of the -1 term in R_e? It seems to just subtract a constant 1 from the regularization term.”
>
> You are right. It provides a simpler formulation in the primal-dual relationship Eq. (7) (this is also used in Genevay et al (2016), Peyré (2016)).

---

> > ### Comment · AnonReviewer4 · 2018-01-03
> > **Clarify relationship**
> >
> > Thanks for your replies.
> >
> > I would strongly recommend clarifying the relationship to the full-dual case of Genevay et al. in the paper. To a reader not intimately familiar with the previous work, it reads as if your dual formulation is also novel, whereas really you're mainly proposing replacing the RKHS with a neural network (and so achieving much better results).

---

> > > ### Author Response · Authors · 2018-01-05
> > > **Agreed**
> > >
> > > We have added the following remark in the paragraph "Algorithm" of page 4 of the updated submission:
> > > "Genevay et al. (2016) used the same stochastic dual maximization approach to compute the regularized OT objective in the continuous-continuous setting. The difference lies in their pamaterization of the dual variables as kernel expansions, while we decide to use deep neural networks."
> > >
> > > We also discussed their cost per iteration, compared to ours, in the paragraph "
> > > Convergence rates and computational cost comparison." when referring to the continuous-continuous setting.
> > >
> > > Thank you for pointing this out.

---

### Official Review · AnonReviewer3 · 2017-12-05
**Very strong paper with novel and interesting results presented clearly and engagingly.**

**Rating:** 8
**Confidence:** 3

**Review:**

This paper explores a new approach to optimal transport. Contributions include a new dual-based algorithm for the fundamental task of computing an optimal transport coupling, the ability to deal with continuous distributions tractably by using a neural net to parameterize the functions which occur in the dual formulation, learning a Monge map parameterized by a neural net allowing extremely tractable mapping of samples from one distribution to another, and a plethora of supporting theoretical results. The paper presents significant, novel work in a straightforward, clear and engaging way. It represents an elegant combination of ideas, and a well-rounded combination of theory and experiments.

I should mention that I'm not sufficiently familiar with the optimal transport literature to verify the detailed claims about where the proposed dual-based algorithm stands in relation to existing algorithms.

Major comments:

No major flaws. The introduction is particular well written, as an extremely clear and succinct introduction to optimal transport.

Minor comments:

In the introduction, for VAEs, it's not the case that f(X) matches the target distribution. There are two levels of sampling: of the latent X and of the observed value given the latent. The second step of sampling is ignored in the description of VAEs in the first paragraph.

In the comparison to previous work, please explicitly mention the EMD algorithm, since it's used in the experiments.

It would've been nice to see an experimental comparison to the algorithm proposed by Arjovsky et al. (2017), since this is mentioned favorably in the introduction.

In (3), R is not defined. Suggest adding a forward reference to (5).

In section 3.1, it would be helpful to cite a reference to support the form of dual problem.

Perhaps the authors have just done a good job of laying the groundwork, but the dual-based approach proposed in section 3.1 seems quite natural. Is there any reason this sort of approach wasn't used previously, even though this vein of thinking was being explored for example in the semi-dual algorithm? If so, it would interesting to highlight the key obstacles that a naive dual-based approach would encounter and how these are overcome.

In algorithm 1, it is confusing to use u to mean both the parameters of the neural net and the function represented by the neural net.

There are many terms in R_e in (5) which appear to have no effect on optimization, such as a(x) and b(y) in the denominator and "- 1". It seems like R_e boils down to just the entropy.

The definition of F_\epsilon is made unnecessarily confusing by the omission of x and y as arguments.

It would be great to mention very briefly any helpful intuition as to why F_\epsilon and H_\epsilon have the forms they do.

In the discussion of Table 1, it would be helpful to spell out the differences between the different Bary proj algorithms, since I would've expected EMD, Sinkhorn and Alg. 1 with R_e to all perform similarly.

In Figure 4 some of the samples are quite non-physical. Is their any helpful intuition about what goes wrong?

What cost is used for generative modeling on MNIST?

For generative modeling on MNIST, "784d vector" is less clear than "784-dimensional vector". The fact that the variable d is equal to 768 is not explicitly stated.

It seems a bit strange to say "The property we gain compared to other generative models is that our generator is a nearly optimal map w.r.t. this cost" as if this was an advantage of the proposed method, since arguably there isn't a really natural cost in the generative modeling case (unlike in the domain adaptation case); the latent variable seems kind of conceptually distinct from observation space.

Appendix A isn't referred to from the main text as far as I could tell. Just merge it into the main text?

---

> ### Author Response · Authors · 2017-12-23
> **Paper updated according to your comments**
>
> Dear reviewer,
>
> Thank you very much for your positive review and detailed comments. Please find below our replies to your comments.
>
> "In the introduction, for VAEs, it's not the case that f(X) matches the target distribution. [...] The second step of sampling is ignored in the description of VAEs in the first paragraph."
>
> At training time, there are indeed two neural networks involved in the VAE model, one encoder and one decoder. Here, we refer to X as the latent variable, i.e. the distribution obtained by the image of the input data by the encoder. We hence refer to f as the decoder network. With these notations, we believe that f is learned so that f(X) matches the distribution of the input data.
>
> "In the comparison to previous work, please explicitly mention the EMD algorithm, since it's used in the experiments."
>
> We used a c++ implementation of the network simplex algorithm (http://liris.cnrs.fr/~nbonneel/FastTransport/). We have added this link as a footnote.
>
> "It would've been nice to see an experimental comparison to the algorithm proposed by Arjovsky et al. (2017), since this is mentioned favorably in the introduction."
>
> Our algorithm shares indeed similarities with the one proposed by Arjovsky et al. (2017): they both use NN parameterizations of the OT dual variables. Both our algorithms have the same complexity. However, in our case, we compute regularized OT, while Arjovsky et al. (2017) unregularized OT. Hence, we found it more relevant to compare to Genevay et al. (2016) who computed exactly the same objective as us (in the entropy reg. case).
>
> "Is there any reason this sort of approach wasn't used previously, even though this vein of thinking was being explored for example in the semi-dual algorithm?"
>
> Let us emphasize that the simplex algorithm is an efficient OT solver for measures supported up to a few thousands samples, and may be suitable in many applications.
>
> It seems that the need to compute OT in large-scale settings is largely driven by the machine-learning community, with the recent idea that the OT objective can be a powerful loss function (Rolet et al. (2016), Arjovsky et al. (2017)), as well as the OT plans can be used to perform domain adaptation (Courty et al. (2016).
>
> Moreover, our dual approach is simple thanks to the convex regularization of the primal OT problem, which was also introduced relatively recently (Cuturi (2013)).
>
> Finally, our approach is not flawless: the use of deep NN makes the problem non-convex (in the semi-discrete and continuous-continuous cases).
>
> "In algorithm 1, it is confusing to use u to mean both the parameters of the neural net and the function represented by the neural net."
>
> We wanted to emphasize that our algorithm is conceptually the same in all settings (discrete-discrete, semi-discrete and continuous-continuous). We are thinking of better notations to make it less confusing.
>
> "There are many terms in R_e in (5) which appear to have no effect on optimization, such as a(x) and b(y) in the denominator and "- 1". It seems like R_e boils down to just the entropy."
>
> We have removed a and b from the text. We can remove the -1 in the entropy regularizer, but 1) it would make the primal-dual relationship less ‘simple’ and 2) this would not be in line with the work of Genevay et al. (2016) or Peyré (2016).
>
> "I would've expected EMD, Sinkhorn and Alg. 1 with R_e to all perform similarly."
>
> We had a typo in the result of the “Bar. proj. Alg. 1 R_e” case. We have rerun the experiment and it indeed performs similarly as Sinkhorn, as expected. We apologize for that.
> EMD (i.e. non-regularized OT) is not expected to perform as Sinkhorn since the regularization has an effect of the OT plan and hence on the barycentric projection.
>
> "In Figure 4 some of the samples are quite non-physical. Is their any helpful intuition about what goes wrong?"
>
> The barycentric projection performs an averaging (w.r.t. the squared Euclidean metric which was chosen in that xp) between target samples (weighted according to the optimal plan). In some case, this averaging might lead to so these non-physical shapes.
>
> "What cost is used for generative modeling on MNIST?"
>
> We used the squared Euclidean distance.
>
> "It seems a bit strange to say "The property we gain compared to other generative models is that our generator is a nearly optimal map w.r.t. this cost" [...] the latent variable seems kind of conceptually distinct from observation space."
>
> Indeed, most generative models do not need to have an 'optimal' the generator. Yet we believe that looking for map which has certain (regularity) properties can be useful for some applications. We may consider further work in generative modeling where optimality of the mapping (w.r.t. to a given cost) can be important (such as image-to-image translation).
>
> References:
> Peyré, Gabriel. "Entropic approximation of Wasserstein gradient flows." SIAM Journal on Imaging Sciences 8.4 (2015): 2323-2351.

---

### Author Response · Authors · 2018-01-05
**Dear reviewers,**

We would like to thank you again for your in-depth review of our submission. Your detailed comments and questions have helped us improve the manuscript and we hope the updated version fulfills your recommendations.

In this research, we have tackled a complicated and open problem which is the computation of optimal transport and optimal mappings in high dimensions for measures on a large (or even continuous) support. This has been possible through the recent developments of optimal transport theory  and machine learning techniques.

We have carefully read and acknowledge your remarks but are not able to positively reply to all, and may keep if as future work:
- First, about the Algorithm 1, we agree that a more thorough numerical convergence analysis on more datasets would be of great interest. After careful thinking, we believe this rigorous analysis requires implementations in a similar framework as well as a variety of datasets that encompasses several ground space dimensions and sizes of distributions. We choose to postpone this study to further analysis and works, as it is not in our opinion the main direction of our paper, which rather focus on the learning of optimal maps.
- Second, we understand convergence rates of discrete regularized OT plans would be a nice addition to our convergence results. As mentioned in the discussion, convergence of discrete OT plan (and not only the OT objective) has not been studied, to our knowledge, in the literature. We believe our convergence results are a first step in that direction.

Let us thank you again for your careful reviewing and interesting discussion.

Best wishes,
The authors.

---

### Decision · Program_Chairs · 2018-01-29
**ICLR 2018 Conference Acceptance Decision**

**Decision:**

Accept (Poster)

**Comment:**

This paper is generally very strong. I do find myself agreeing with the last reviewer though, that tuning hyperparameters on the test set should not be done, even if others have done it in the past. (I say this having worked on similar problems myself.) I would strongly encourage the authors to re-do their experiments with a better tuning regime.